# Tea Tree Essential Oil Kills *Escherichia coli* and *Staphylococcus epidermidis* Persisters

**DOI:** 10.3390/biom13091404

**Published:** 2023-09-18

**Authors:** LeeAnn Nguyen, Brianna DeVico, Maliha Mannan, Matthew Chang, Cristina Rada Santacruz, Christopher Siragusa, Sydney Everhart, Christopher H. Fazen

**Affiliations:** Department of Chemistry, Drew University, Madison, NJ 07940, USA; lnguyen1@drew.edu (L.N.); bdevico@drew.edu (B.D.); mmannan@drew.edu (M.M.); mchang1@drew.edu (M.C.); cradasantacruz@drew.edu (C.R.S.); csiragusa@drew.edu (C.S.); severhart@drew.edu (S.E.)

**Keywords:** bacterial persisters, essential oils, tea tree essential oil

## Abstract

Persister cells are a small subpopulation of non-growing bacteria within a population that can survive long exposures to antibiotic treatment. Following antibiotic removal, persister cells can regrow and populate, playing a key role in the chronic reoccurrence of bacterial infections. The development of new molecules and methods to kill bacterial persisters is critical. Essential oils and other natural products have long been studied for their antimicrobial effects. Here, we studied the effectiveness of tea tree essential oil (TTO), a common component in many commercial care products, against *Escherichia coli* and *Staphylococcus epidermidis* persister cells. Using biphasic kill curve assays, we found that concentrations of 0.5% and 1.0% TTO for *E. coli* and *S. epidermidis*, respectively, completely eradicated persister cells over a period of 24 h, with the component terpinen-4-ol responsible for most of the killing. Using a colorimetric assay, it was determined that the TTO exhibited its anti-persister effects through a membrane disruption mechanism.

## 1. Introduction

Antibiotic resistance has been a worsening problem since the very introduction of widespread antibiotic usage. In 2019, the Centers for Disease Control and Prevention (CDC) released an Antibiotic Resistance (AR) report that stated that 2.8 million people are impacted by antibiotic resistant infections, in which 48,000 people die annually. This leads to increasing difficulties in the treatment of bacterial infections, as the newly developed drugs are often less effective, more toxic, and more expensive [1].

In addition to the development of resistance, bacterial populations contain a percentage of cells that are tolerant to antimicrobials called persister cells. First discovered in the early 1940s [2], these cells are a phenotypic variant that are metabolically inactive, and as such, are able to evade death from conventional antibiotics that target metabolically active bacteria [3,4]. In response to environmental stress, these persister cells enter a dormant state, and although they cannot grow, they can survive antibiotic treatment [5,6,7,8]. When the antibiotic is removed, the persister cells “awaken”, leading to growth and chronic reinfection [3,9,10,11]. Further, research has suggested a potential link between antibiotic tolerance in persisters and antibiotic resistance [12,13].

While some methods and molecules for persister cell eradication have been developed in recent years [14,15,16,17,18,19], in order to withstand the rapid growing rate of antibiotic resistance within bacteria and microorganisms, the focus has turned toward the development and research of new antimicrobial agents and resistance modifiers. An expanding area of study looks at the utilization of medical plants that naturally contain properties that can act as alternative antibacterial agents [20,21,22]. These natural products have novel mechanisms of action that are not only effective but also mitigate side effects associated with antimicrobials. One common mechanism of action is through membrane disruption [23,24,25]. This mode of action is particularly appealing in combatting persisters cells that are in a metabolically inactive state.

In this study, we looked at tea tree essential oil (TTO), a naturally derived mixture found commonly in personal care products. TTO has been shown to have antimicrobial properties and to exert those properties through inhibition of respiration and increasing of membrane permeability [26,27]. We hypothesized that because of these characteristics, TTO could kill persister cells. Here, we show that TTO is a broad-spectrum anti-persister agent that effectively kills persister cells from both Gram-negative (*Escherichia coli*) and Gram-positive (*Staphylococcus epidermidis*) bacteria.

## 2. Materials and Methods

### 2.1. Bacterial Strains, Chemicals, and Growth Media

The strains used in this study, *Escherichia coli* MG1655 and *Staphylococcus epidermidis* RP62A were acquired from the American Type Culture Collection (ATCC; Manassas, VA, USA). Tea tree essential oil (TTO) was purchased from Plant Therapy (Twin Falls, ID, USA). A certificate of analysis and gas chromatography–mass spectrometry (GC–MS) analysis was provided by the supplier. The full chemical composition of the TTO is provided in the Appendix A. A 70% *v*/*v* TTO solution was prepared in dimethyl sulfoxide (DMSO; Sigma-Aldrich, St. Louis, MO, USA) and filter sterilized with a 0.22 μm syringe filter the same day as the experiment. Individual TTO components terpinen-4-ol, α-terpinene, and γ-terpinene were purchased from Sigma-Aldrich.

Media components tryptic soy broth (TSB), tryptone, yeast extract, agar, NaCl, KCl, Na_2_HPO_4_, and KH_2_PO_4_ were purchased from Thermo Fisher Scientific (Waltham, MA, USA). For *E. coli*, lysogeny broth (LB; 10 g/L tryptone, 5 g/L yeast extract, and 10 g/L NaCl) was used for planktonic growth, and LB-agar (LB + 15 g/L agar) was used to measure colony forming units (CFUs). For *S. epidermidis*, TSB was used for planktonic growth, and TSB-agar (TSB + 15 g/L agar) was used to measure CFUs.

Erythrosin B (EB), NaOH, HCl, and the antibiotics ofloxacin and ciprofloxacin were purchased from Sigma-Aldrich. Stock solutions of ofloxacin and ciprofloxacin were prepared in 0.1 M NaOH and 0.1 M HCl, respectively, diluted in ultrapure water, and filter sterilized with a 0.22 μm syringe filter the same day as the experiment.

### 2.2. Determining Minimum Inhibitory Concentrations

The minimum inhibitory concentrations (MICs) for TTO against *E. coli* and *S. epidermidis* were determined by a broth dilution method described previously [28]. Briefly, test tubes were inoculated with 10^5^ CFU/mL of *E. coli* (in LB) or *S. epidermidis* (in TSB) supplemented with TTO at the following concentrations: 0, 0.0156%, 0.0313%, 0.0625%, 0.125%, 0.25%, 0.5%, 1%, 2%, 4%, and 8%. The TTO concentrations were achieved through the dilution of a filter-sterilized 70% *v*/*v* TTO/DMSO solution with LB or TSB. The test tubes (final volume of 1 mL) were incubated statically at 37 °C for 24 h. The MIC was determined the lowest concentration of TTO for which no visible bacterial growth was observed.

### 2.3. Obtaining Persister Cells

LB (or TSB) media (5 mL) was inoculated with *E. coli* (or *S. epidermidis*) from a single aliquot –80 °C stock and incubated for 4 h at 37 °C, with shaking at 250 rpm in a test tube. An overnight culture was made with 250 µL of the starter culture and 25 mL of LB (or TSB) media in a 250 mL baffled Erlenmeyer flask and was incubated for 18 h at 37 °C, with shaking at 250 rpm. To obtain persister cells, the 25 mL overnight culture was treated as follows: for *E. coli*, 24 h at 37 °C, 250 rpm, with 5 μg/mL ofloxacin; for *S. epidermidis*, 48 h at 37 °C, 250 rpm, with 10 μg/mL ciprofloxacin. After the treatment period, the culture was centrifuged at 4000× *g* for 10 min, the supernatant was removed, and the pellet was resuspended in 25 mL of phosphate-buffered saline (PBS; 137 mM NaCl, 2.7 mM KCl, 10 mM Na_2_HPO_4_, and 1.8 mM KH_2_PO_4_ at pH 7.4). It was determined that persisters were formed under these treatment conditions by generation of time-dependent kill curves [29], see Appendix A.

### 2.4. Tea Tree Oil and Components Killing Assays

Aliquots of the persister cells in PBS were combined with varying concentrations of TTO, TTO components (terpinen-4-ol, α-terpinene, and γ-terpinene), or DMSO (solvent control at the highest percentage used in the TTO samples) to a final volume of 1 mL in test tubes. Samples were incubated at 37 °C, with shaking at 250 rpm for 1, 2, and 24 h. At each time point, samples were removed, centrifuged, washed 3× in PBS (to drop concentration of TTO/component/DMSO below MIC), and resuspended in PBS. The samples were serially diluted in PBS, and 10 μL spots were plated on LB-agar (*E. coli*) or TSB-agar (*S. epidermidis*). Following static incubation for 16 h at 37 °C, persisters were enumerated by counting CFUs. Colony counts between 10 and 100 were used.

### 2.5. Assessment of Membrane Integrity by Erythrosin B Assay

Following treatment of persister cell fractions with TTO, the TTO components, or DMSO, to determine the membrane integrity of the cells, an erythrosin B (EB) assay was performed, as previously described [30]. Briefly, 500 μL samples were combined 1:1 with either PBS or 0.08% EB dye (prepared in PBS) and incubated for 15 min. After incubation, the samples were centrifuged (10,000× *g* for 1 min), the supernatant was removed, and the samples were resuspended in 1 mL of PBS. Absorbance of the EB dye was measured at 530 nm. Data were analyzed by measuring the amount of EB absorbance above the control (no treatment) sample.

### 2.6. Statistical Analysis

All experiments were performed with at least three biological replicates. For each time point assessed, mean values and the standard error of the mean were calculated. For analysis of statistical significance, a two-tailed Student’s *t*-test was used versus the control conditions. Data considered significant have *p*-values less than 0.05.

## 3. Results

### 3.1. Tea Tree Essential Oil Kills E. coli and S. epidermidis Persister Cells

Persister cells were acquired through the treatment of stationary-phase cultures with antibiotic concentrations in excess of the MIC values until a biphasic killing curve was obtained (Appendix A). In order to determine test concentrations for TTO, we measured the TTO MIC values for *E. coli* and *S. epidermidis*. Using a broth dilution method, we obtained MIC values of 0.5% and 1% TTO for *E. coli* and *S. epidermidis*, respectively. These MIC values were used as starting points in our persister killing assays.

After isolating persister cells, the cultures were treated with varying concentrations of TTO starting at 1× MIC. As shown in Figure 1a, all concentrations of TTO tested completely eradicated all *E. coli* persister cells within 1 h of treatment. Because DMSO was used as a solubilizing vehicle for the TTO, it was also tested. There was no significant persister cell killing in the samples treated with DMSO. TTO also showed significant (*p* < 0.05) killing in *S. epidermidis* (Figure 1b). Unlike in the *E. coli* sample, there were detectable persisters after 1 and 2 h in the *S. epidermidis* sample; however, complete killing of persisters was observed after 24 h of treatment (Figure 1b).

### 3.2. Terpinen-4-ol Is the Primary Killing Agent in Tea Tree Essential Oil

TTO is a mixture comprising several components (see Appendix A). After we determined that TTO was effective in killing *E. coli* and *S. epidermidis* persister cells, we wanted to determine which component or components were responsible for the killing. We looked at the three components of TTO that make up the largest percentages of the oil: terpinen-4-ol (42.59% of TTO), α-terpinene (9.12% of TTO), and γ-terpinene (20.10% of TTO). As shown in Figure 2a–c, terpinen-4-ol significantly (*p* < 0.05) kills most of the *E. coli* persister cells. α-Terpinene shows that it might contribute to the anti-persister effect to a small degree. For *S. epidermidis* (Figure 2d–f), terpinen-4-ol is shown again to be the primary component exerting the anti-persister effects of TTO, completely eradicating persister cells after just 1 h of treatment. As seen in Figure 2e, after 24 h of treatment, α-terpinene also significantly (*p* < 0.05) kills a large percentage of persisters, although not to the same effect as terpinen-4-ol.

### 3.3. Tea Tree Essential Oil Disrupts the Persister Cell Membrane

Having established that TTO kills persister cells, we set out to explore a possible mechanism of action. Given what is known regarding how TTO exerts its effects on non-persister bacteria [26], we hypothesized that TTO acts by disrupting the persister cell membrane. To explore this, we performed a colorimetric assay using erythrosin B (EB). EB has been recently shown to be useful as a membrane-exclusion dye for the study of live–dead bacteria [30]. Bacteria with a compromised membrane will appear as dye positive. In order to determine if persister cell death was due to a disrupted membrane, cells were incubated with EB following treatment with TTO. As can be seen in Figure 3, the absorbance at 530 nm (due to the EB dye) for the TTO-treated samples was significantly (*p* < 0.05) larger than the absorbance of the non-treated (PBS) cells or those cells that were treated with DMSO (as a vehicle control). This result was observed for both the *E. coli* cells (Figure 3a) and for the *S. epidermidis* cells (Figure 3b). The DMSO controls were not significantly different than the no-treatment (PBS) controls, indicating that the cell membrane disruption was due to the TTO and not the DMSO in which it was solubilized.

## 4. Discussion and Conclusions

As bacteria continue to develop antibiotic resistance, there is a pressing need for the development of new antibiotics or methods for killing these organisms. Owing to their ability to survive high concentrations of antibiotic, persister cells also play a key role in reoccurring and prolonging bacterial infections [3,9,10,11]. In addition, there is evidence that persisters play a role in the development of antibiotic resistance [12,13]. In spite of this, many pharmaceutical companies are no longer looking to develop new antibiotics. As of 2017, the estimated cost to develop a new antibiotic was USD 1.5 billion. With an average return on that investment of only about USD 46 million per year, antibiotic development does not provide a good return on investment [31]. To that end, attention has been turned to the repurposing currently approved drugs [15] or the investigation of natural products, such as essential oils [20,21,22].

In this study, we investigated the effect of the essential oil TTO. Although the antibacterial properties of TTO against antibiotic-susceptible non-persister bacteria have been known for some time [26,27], we wanted to explore the effects of TTO against bacterial persisters. Could TTO eradicate these persisters that characteristically survive and tolerate high concentrations of conventional antibiotics? Further, as broad-spectrum anti-persister agents are of greater use, we performed our tests against Gram-negative (*E. coli*) and Gram-positive (*S. epidermidis*) bacteria. Gram-negative bacteria have an outer membrane; Gram-positive bacteria lack an outer membrane but have a much thicker peptidoglycan layer outside their inner membrane. TTO was able to completely eradicate persisters of both *E. coli* and *S. epidermidis* within 1 and 24 h, respectively. Although TTO consists of several components, we showed that it was terpinen-4-ol that was responsible for wiping out *E. coli* and *S. epidermidis* at concentrations of 0.5% and 1.0%, respectively. This is not surprising given that past research into the antimicrobial effects of TTO on non-persister cells has shown terpinen-4-ol exhibits the primary effect [21,32,33].

Finally, persister survival results from the fact that common conventional antibiotics target ATP-dependent processes, such as protein synthesis (aminoglycosides), DNA replication (fluoroquinolones), and cell wall synthesis (β-lactams). Persister cells that exist in a dormant state have been shown to have lower levels of ATP compared to bacteria that are metabolically active, and as such, are able to evade killing from these conventional antibiotics [34]. In order to kill persisters, our targets need to be non-ATP-dependent. Attacking the cell membrane provides a mode of action that could allow us to kill persisters while they remain in their dormant state [35]. Here, we utilized a colorimetric assay to show that TTO did in fact disrupt the cell membranes of both *E. coli* and *S. epidermidis* resulting in cell death. Overall, these findings provide evidence for the potential usage of TTO as an anti-persister therapeutic.

## Figures and Tables

**Figure 1 biomolecules-13-01404-f001:**
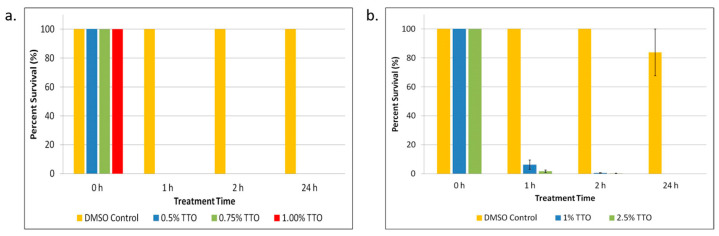
Tea tree essential oil (TTO) treatment of *E. coli* and *S. epidermidis* persister cells. (**a**) *E. coli* persister cells and (**b**) *S. epidermidis* persister cells were treated with different concentrations of TTO over a period of 1, 2, and 24 h. After treatment, persister cells were determined. Percent survival against an untreated control was plotted. Data points represent three replicate experiments, and error bars represent standard error of the mean. DMSO control is the solvent vehicle control for TTO at the highest concentration used for TTO treatment.

**Figure 2 biomolecules-13-01404-f002:**
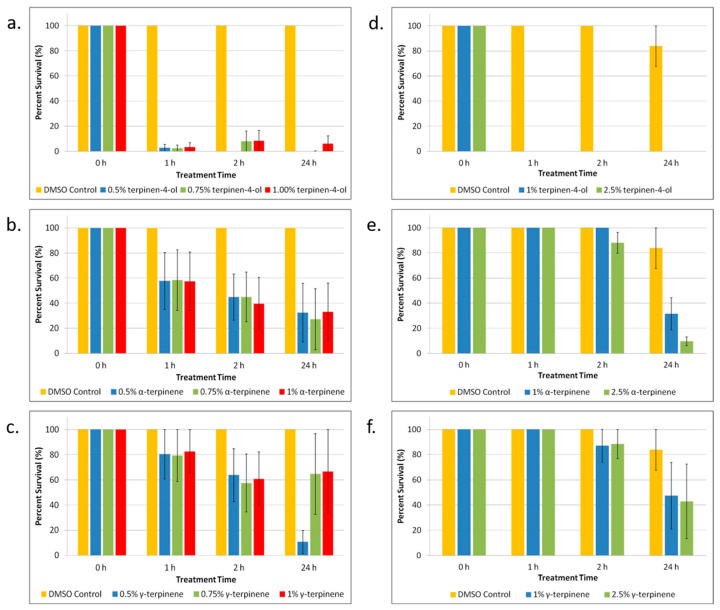
Tea tree essential oil (TTO) component treatment of *E. coli* and *S. epidermidis* persister cells. (**a**–**c**) *E. coli* persister cells and (**d**–**f**) *S. epidermidis* persister cells were treated over a period of 1, 2, and 24 h with different concentrations of the three primary components of TTO: (**a**,**d**) terpinen-4-ol; (**b**,**e**) α-terpinene; and (**c**,**f**) γ-terpinene. After treatment, persister cells were determined. Percent survival against an untreated control was plotted. Data points represent three replicate experiments, and error bars represent standard error of the mean. DMSO control is the solvent vehicle control for TTO at the highest concentration used for TTO treatment.

**Figure 3 biomolecules-13-01404-f003:**
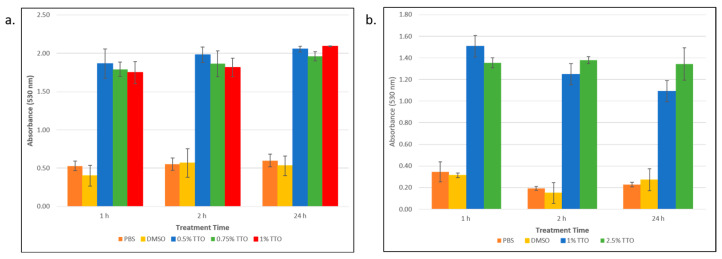
Determining the membrane integrity of *E. coli* and *S. epidermidis* persister cells following treatment with tea tree essential oil (TTO). (**a**) *E. coli* persister cells and (**b**) *S. epidermidis* persisters were treated with different concentrations of TTO over a period of 1, 2, and 24 h. Samples were incubated with erythrosin B dye, and following washes and centrifugation steps, the sample absorbance was measured at 530 nm. This absorbance was corrected for background cell absorption and plotted against treatment time. Data points represent three replicates, and error bars represent standard error of the mean. DMSO control is the solvent vehicle control for TTO at the highest concentration used for TTO treatment.

## Data Availability

All data are available upon request.

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
