# Peer review of "Tea Tree Essential Oil Kills Escherichia coli and Staphylococcus epidermidis Persisters"

_biomolecules, 2023, doi:10.3390/biom13091404_

Round 1

Reviewer 1 Report

The manuscript “Tea Tree Oil Kills Escherichia coli and Staphylococcus epidermidis Persisters” describes the effect of tea tree essential oil (EO) on Escherichia coli and Staphylococcus epidermidis persisters cells. The article provides information regarding the effect of tea tree EO as well as its main components against Escherichia coli and Staphylococcus epidermidis persistent cells. Authors determined MIC, obtained persister cells and determined membrane integrity of EO and its main components. The overall scientific soundness of the paper is solid, English is very good and some interesting findings are presented.

I have only few minor observations

-        The title states oil, instead of essential oil. Please correct and accentuate in the entire manuscript.

-        Line 60 All other – please specify

The manuscript “Tea Tree Oil Kills Escherichia coli and Staphylococcus epidermidis Persisters” describes the effect of tea tree essential oil (EO) on Escherichia coli and Staphylococcus epidermidis persisters cells. The article provides information regarding the effect of tea tree EO as well as its main components against Escherichia coli and Staphylococcus epidermidis persistent cells. Authors determined MIC, obtained persister cells and determined membrane integrity of EO and its main components. The overall scientific soundness of the paper is solid, English is very good and some interesting findings are presented.

I have only few minor observations

 -        The title states oil, instead of essential oil. Please correct and accentuate in the entire manuscript.

-        Line 60 All other – please specify

Reviewer 2 Report

The article entitled „Tea Tree Oil Kills Escherichia coli and Staphylococcus epidermidis Persisters” concern vivid and valid problem of antibiotic resistance and searching for new molecules to combat the problem. In the article tea tree essential oil was used to kill E. coli and S. epidermis persister cells. The introduction is informative and enough present current state of the problem. The research was planned properly with adequate methods used. The article itself is clear, well prepared (shortly and clearly presenting the problem described). However some corrections should be done, including minor - e.g. some abbreviations are not explained (CDC – line23); and major – in the article there is a lack of chemical analysis of the essential oil – its composition and content of at least dominants should be determined. The content and composition of essential oil in plant raw materials (of the same species) can vary thoroughly depending on many factors (e.g. genetic diversity of the species, age of the plant, term of harvest, postharvest treatment of raw materials and others), that’s why these parameters in such type of experiment should be presented (especially when the authors purchase the product from a company - oftenly such essential oils are modified by adding some naturally present compound to sthrenthen activity of the oil).

Round 2

Reviewer 2 Report

Proper corrections have been done.